# Nutritional and Physiological Demands Shape the Gut Microbiome of Female World Tour Cyclists

**DOI:** 10.3390/microorganisms13102345

**Published:** 2025-10-13

**Authors:** Toon Ampe, Lieselot Decroix, Kevin De Pauw, Romain Meeusen, Thomas Demuyser, Bart Roelands

**Affiliations:** 1Human Physiology and Sports Physiotherapy Research Group (MFYS), Vrije Universiteit Brussel (VUB), 1050 Brussels, Belgium; toon.ampe@vub.be (T.A.);; 2Brussels Human Robotics Research Centre, BruBotics, Vrije Universiteit Brussel (VUB), 1050 Brussels, Belgium; 3Department of Sports, Recreation, Exercise and Sciences [SRES], Faculty of Community and Health Sciences, University of the Western Cape, Cape Town 7535, South Africa; 4Department of Microbiology, Universitair Ziekenhuis Antwerpen (UZA), 2650 Edegem, Belgium; 5AIMS Laboratory, Centre for Neurosciences, Faculty of Medicine and Pharmacy, Vrije Universiteit Brussel (VUB), 1050 Brussels, Belgium; 6Laboratory of Applied Microbiology and Biotechnology, Department of Bioscience Engineering, University of Antwerp, 2000 Antwerp, Belgium; 7Laboratory of Sports and Nutrition Research, Riga Stradins University, LV-1007 Riga, Latvia

**Keywords:** female cyclists, gut microbiome, nutrition, professional cycling, cycling performance, gut health

## Abstract

This cross-sectional study investigated whether elite female World Tour cyclists have a specific gut microbiome compared to non-athlete female controls, potentially resulting from the unique physiological and dietary demands of high-level endurance cycling. Fourteen female cyclists and thirteen matched controls provided fecal samples during a period of reduced training (off-season cycling). The samples were analyzed using 16S rRNA gene sequencing and short-chain fatty acid (SCFA) quantification. The results revealed significant differences in microbiome composition. The cyclists showed a higher abundance of Bacteroidota (72.7% vs. 15.3%) and a lower abundance of Firmicutes (22.1% vs. 62.5%) compared to the controls, along with reduced alpha-diversity (Shannon index, *p* < 0.05). Fiber-fermenting families such as *Lachnospiraceae* and *Ruminococcaceae* were depleted, consistent with a carbohydrate-focused and relatively low-fiber diet. Interestingly, fecal SCFA levels did not differ, suggesting functional adaptation of the microbiome. These findings indicate that the elite female cyclists may have developed a “performance-adapted” gut microbiome. However, due to the cross-sectional design, causality cannot be established, and the long-term health implications remain uncertain.

## 1. Introduction

The gastrointestinal tract is home to trillions of microorganisms, including bacteria, viruses, fungi, and archaea [1], which make up a complex ecological community with versatile functions, like energy production, regulating the immune system, producing hormones, and communicating with the brain [2,3,4,5]. An important mechanism linking the gut microbiota to host health and performance lies in their fermentation of nondigestible carbohydrates into short-chain fatty acids (SCFAs), particularly acetate, propionate, and butyrate [6,7,8]. The intestinal epithelium extensively oxidizes butyrate, contributing to intestinal health via localized effects [9]. Acetate and propionate are also utilized in terms of oxidation, although to a lesser extent than butyrate [10]. SCFAs that are not oxidized can either feed into the Krebs cycle as substrates or enter systemic circulation, causing positive effects in various cells and organs [11]. These metabolites contribute to approximately 10% of the host’s daily caloric requirements. Additionally, the gut microbiota has a direct or indirect impact on factors like appetite, fat storage, glucose tolerance, the immune system, and the gut–brain axis [12,13,14,15].

Recently, exercise has been found to be an effective modulator in both the composition and metabolic activity of the gut microbiota [16]. The literature on the effects of chronic exercise on the gut microbiome showed positive effects on alpha-diversity, compared to non-athletes, which is linked to cardiorespiratory fitness (i.e., VO_2_ max) [17]. Alongside this, positive effects on specific, health-promoting, taxa such as *Akkermansia*, *Bifidobacterium*, and *Lactobacillus* spp. were found [18,19,20,21]. However, the relationship between exercise and the microbiome is complex. While moderate exercise enhances gut health by promoting beneficial oxygen conditions for butyrate-producing microbes, prolonged intense exercise decreases splanchnic blood flow, causing gut ischemia, increased permeability, and microbiota dysbiosis [22,23,24,25,26]. Interestingly, elite endurance athletes often have a reduced gut microbial diversity combined with elevated SCFA concentrations. This suggests a functional specialization of their microbiome to support high-energy demands and metabolic resilience [27].

Additionally, an altered diet is often found in an athletic population, especially during competition and intensified training periods. Since diet is the most important factor determining the gut microbiome composition, those sport/athlete-specific diets might lead to an altered gut microbiome with specific taxa and changes in diversity [24]. The effect of macro- and micronutrients in healthy and patient populations is already described extensively in the literature [28]. However, the effect of the interplay between exercise and diet on the gut microbiome in an athlete population has not yet been examined. According to the International Olympic Committee (IOC) consensus on sports nutrition, the endurance athlete’s diet typically consists of low-fat, low-protein, and high-carbohydrate intake before and during the effort [29]. In the recovery phase, a higher protein intake is advised. For carbohydrate intake, complex and indigestible starches and fibers are limited due to their undesirable properties in a sport context, like water retention, impaired absorption of minerals, and gastrointestinal distress. Furthermore, evidence suggests that the interaction between high-intensity training and athlete-specific diets may also induce low-grade metabolic acidosis, further influencing gut microbiome and systemic health [30].

Emerging from preliminary observations in both animal and human models, interventions that focus on optimizing microbial balance (such as probiotic, prebiotic, and SCFA supplementation) have shown (in-)direct improvements in performance [31]. These results highlight the ergogenic potential associated with the gut microbiota. For instance, Bacteroides species, which thrive on simple carbohydrates, dominate in athletes and may enhance substrate metabolism, while depletion of fiber-fermenting taxa (e.g., *Lachnospiraceae*) reflects dietary strategies to minimize gastrointestinal symptoms during competition [13,32]. Notably, such dietary patterns can alter colonic pH gradients, favoring proteolytic fermentation and potentially compromising gut barrier integrity [13,30,33].

Despite this research, significant knowledge gaps remain, particularly in elite (female) athletes. Women are underrepresented in sports science research, and sex-specific physiological and hormonal factors may lead to unique microbiome adaptations [34,35]. This study aimed to address this gap by profiling the gut microbiome of elite female cyclists competing at the World Tour level. It is hypothesized that their gut microbiome has evolved due to the chronic physical stressor of cycling at the highest level and the specific diet and supplementation used to optimize their endurance performance. Therefore, to our knowledge, this was the first study to comprehensively profile the gut microbiome, using both 16S rRNA sequencing and SCFA quantification, in elite female World Tour cyclists during the off-season, comparing them to well-matched non-athlete controls.

## 2. Materials and Methods

### 2.1. Participants

Fifteen female cyclists from a single UCI World Tour team were recruited through purposive sampling for this study. The eligibility criteria included sports medical approval, absence of inflammatory bowel disorder or similar conditions, no alcohol or other substance abuse, no smoking, and no anti- or probiotic use in the two weeks prior to participation. Information on other medications and supplements was provided by the cyclists to the team dietitian and/or team doctor. The control group (n = 15) consisted of age-matched (i.e., age range of the cyclists: 18–34 years) and BMI-matched (<25 kg/m^2^) females. Furthermore, to focus on the effects of elite cycling on the gut microbiome, controls were excluded if they were endurance-trained (International Physical Activity Questionnaire (IPAQ) categories low-moderate only). All the other eligibility criteria were applied equally to both groups. The participants who fulfilled the predefined criteria were recruited from connections of the involved researchers and their research group.

During the off-season (end of October 2024), the participating cyclists were free from training and lifestyle obligations from the team for two to three weeks. In order to take a sample as close as practically possible to the off-season, this baseline measurement took place during the first team meeting at the beginning of December 2024. This sample was used to assess the effects of different phases of a cycling season compared to a relatively untrained state (i.e., longitudinal study). Measurements of the control group took place in the first semester of 2025. Before their participation, all the participants provided informed consent, and the experimental protocol received approval from the local ethics committee (BUN: 1432024000285).

### 2.2. Assessment Procedures

#### 2.2.1. Diet

After providing informed consent, participants from both groups were asked to complete a food diary (incl. supplements) for the two days before taking a fecal sample, to assess the effect of differences in diet on their gut microbiome. A two-day food diary was chosen to minimize participant burden on the elite athletes while still capturing recent dietary intake, acknowledging that it may not fully represent long-term habitual diet. Future studies would benefit from longer tracking periods or food frequency questionnaires. The participants described and/or sent pictures of their diet to the head researcher in a private conversation. Afterward, their food diaries were processed in the Nubel Voedingsplanner (Nubel v.z.w., Brussel, Belgium) to analyze the macro- and micronutrient content of their diet. Averages for both days were calculated.

#### 2.2.2. Physical Activity Data

For the control group, the IPAQ short version was used to assess their habitual physical activity level [36]. The IPAQ is a widely used self-report questionnaire designed to assess physical activity levels in adults aged 15–69 years. It captures data on the frequency and duration of physical activity across three intensity levels: vigorous, moderate, and walking, as well as time spent sitting. The tool provides an estimate of the total physical activity per week and sedentary behavior, allowing classification into low, moderate, or high activity levels.

Training data from the cyclists’ Garmin (Olathe, KS, USA) sport watch (Garmin Forerunner 965), heart rate belt (Garmin HRM-Pro), cycling GPS (Garmin EDGE 1050), and Shimano power meter (Dura-Ace R9200-P, Eindhoven, The Netherlands) were uploaded to their personal TrainingPeaks platform (TrainingPeaks, Louisville, KY, USA). From this platform, training load and training duration were used to calculate the Training Stress Score (TSS) of each training in the week before sampling. In addition, heart rate and power data were extracted to calculate the energy expenditure of their training.

#### 2.2.3. Health Parameters

The prevalence of GI symptoms was assessed using the Bristol Stool Chart (BSC) and the gastrointestinal symptoms short questionnaire (GISS). The BSC is a diagnostic tool used to classify the form of human feces into seven categories. It is used to evaluate the consistency, shape, and frequency of bowel movements, and can be used to identify possible problems such as constipation or diarrhea. Type 1 and 2 indicate (mild) constipation; type 3 and 4 indicate a normal bowel movement; and type 5, 6, and 7 indicate diarrhea [37]. GISS is a tool used to evaluate the severity of symptoms in patients with functional gastrointestinal disorders. The GISS is a self-reported questionnaire that consists of 13 items that assess the severity of the most common GI symptoms on a 10-point scale, where 0 indicates the absence of symptoms and 10 indicates very severe symptoms. To limit the burden on the cyclists, GI symptoms were assessed using a general score on a 10-point scale. Lastly, the participants were asked to report their last menstrual phase and the use of hormonal contraception, along with other medications.

#### 2.2.4. Fecal Samples

During the introduction moment, the participants were provided with a self-sampling kit to collect their feces once, as fast as possible, after the two-day food diary. Full instructions on the sampling technique were given to the participant, and a small folder with this information was provided (FGFP home sampling procedures) [38]. The sampling kit contained a sterilized feces container, eNat-tube with swab (Copan, Brescia, Italy), and feces catcher. After taking the feces sample, it was immediately placed in the freezer of the participant. The sample was kept in the freezer until one of the researchers came to collect it. The samples were frozen at −20 °C until frozen transport (i.e., portable freezer) to the research facility. Upon arrival at the research facility, samples were stored at −80 °C until further analysis.

For the analysis of the fecal samples, 16S rRNA sequencing was performed at the Microbiology and Infection Control department (UZ Brussel, Brussels, Belgium). Microbial DNA was extracted from 10 mg of the fecal samples using the DNeasy PowerSoil Pro Kit (Qiagen, Hilden, Germany), to which 20 μL of ZymoBIOMIC^TM^ (Zymo, Irvine, CA, USA) Spike-in Control I High Microbial Load was added. Appropriate positive and negative controls were included to distinguish relevant signals from contamination or noise. DNA quantification was performed using the Qubit^TM^ dsDNA high sensitivity range assay kit (ThermoFisher Scientific, Waltham, MA, USA). The extracted genetic material was then sequenced by BRIGHTCore (UZ Brussel, Brussels, Belgium), targeting the V4 hypervariable regions of the bacterial 16S rRNA gene and using the NovaSeq platform (Illumina, San Diego, CA, USA). Sequence data generated as FASTQ files were analyzed using the nf-core/ampliseq software 2.13.0 to obtain the taxonomic profile of each sample.

SCFAs (i.e., butyrate, acetate, and propionate) were extracted from the fecal samples by ProDigest (Gent, Belgium) using acetonitrile after acidification with formic acid (0.5% *v*/*v*). 2-Methyl hexanoic acid was added as an internal standard. The extracts were analyzed using a GC-2030 gas chromatograph (Shimadzu, ‘s-Hertogenbosch, The Netherlands), equipped with a SH-PolarD capillary column (30 m × 0.32 mm, film thickness 1.00 µm; Shimadzu, Tokyo, Japan), a flame ionization detector, and a split injector. The injection volume was 1 µL, and the oven temperature was initially set at 110 °C, followed by an increase of 6 °C min^−1^ to a final temperature of 200 °C. Nitrogen was used as the carrier gas, with injector and detector temperatures set at 200 °C.

### 2.3. Statistical Analysis

All the statistical analyses were conducted in R version 2025.05.0+496 (R Core Team, 2013), with statistical significance set at *p* < 0.05 (two-tailed). Descriptive statistics are presented as mean ± standard deviation (SD) or median and interquartile range (IQR), depending on the distribution of the data. Normality was assessed using Shapiro–Wilk tests and visual inspection of Q-Q plots.

Group comparisons (i.e., cyclists vs. controls) for continuous variables (e.g., age, BMI, macronutrient intake, GISS, and BSC scores) were performed using independent samples *t*-tests for normally distributed data or Wilcoxon rank-sum tests for non-normally distributed data. Effect sizes were reported as Cohen’s d or rank-biserial correlation (r), respectively. Gut microbiome α-diversity metrics (i.e., Shannon index, Amplicon Sequence Variants (ASV) richness, and Pielou’s evenness) were compared between the groups using linear regression models (β coefficients with 95% confidence intervals [CI]). β-diversity (Bray–Curtis and Jaccard dissimilarity) was analyzed using PERMANOVA (Adonis test) implemented in the vegan package, with significance assessed via F-statistics and R^2^ values. Differential abundance of bacterial taxa (i.e., phylum and family level) was assessed using Wilcoxon rank-sum tests due to the non-normal distribution of relative abundance data. Log_2_ fold changes (log_2_FC) and effect sizes (r) were calculated for significant taxa. The results were visualized using volcano plots (threshold: *p* < 0.05 and |log_2_FC| > 1). Fecal SCFA concentrations (i.e., acetate, propionate, and butyrate) were compared between the groups using independent samples *t*-tests, with effect sizes reported as Cohen’s d.

## 3. Results

### 3.1. Descriptives

In total, three participants were excluded from the study. One cyclist was excluded because of not participating in the first training camp of the team. Within the control group, two participants were excluded from the analysis due to a BMI higher than 25 kg/m^2^ at the moment of measurement, which is a confounding factor [39]. This resulted in the participation of 14 cyclists and 13 matched controls. The cyclists had an average age of 26.4 ± 4.1 years with a BMI of 20.4 ± 1.3 kg/m^2^, which was not significantly different from the control group (*p* = 0.236, *p* = 0.175, respectively), with an average age of 24.4 ± 4.6 years and a BMI of 21.3 ± 2.1 kg/m^2^.

All the cyclists included were of UCI World Tour level. Their first Lactate Threshold 1 (LT1) was on average at 3.83 ± 0.30 W/kg, and their LT2 at 4.71 ± 0.35 W/kg. The week before the sampling, the cyclists trained 16.4 ± 4.7 h, whereof 13.3 ± 4.1 h on the bicycle, 0.6 ± 0.7 h running, and 2.5 ± 1.1 h strength training or yoga. This resulted in an average total TSS of 687.8 ± 241.8. The control group indicated on the IPAQ to participate on average 3.1 ± 1.5 h/week in moderate and/or heavy exercise, which was significantly lower (*p* < 0.001).

### 3.2. Diet and Gastrointestinal Symptoms

The average active energy expenditure of the cyclists was 1121 kcal/day (IQR = 319), which was associated with a significantly higher energy intake of 2499 kcal (IQR = 187) compared to 1453 kcal (IQR = 660) in the control group (W = 159, *p* = 0.001, r = 0.63). Regarding their relative macronutrient intake, a higher percentage of protein intake was found in the cyclists (17.6%, IQR = 3.2) compared to the matched controls (14.8%, IQR = 2.0) (W = 149, *p* = 0.004, r = 0.54). For relative fat and carbohydrate intake, no significant differences were found between the two groups (see Table 1). Nevertheless, when looking more into detail at the carbohydrate intake, a significantly lower starch intake was found in the cyclists (45.9%, IQR = 13.9) (Med. diff. = −20.4, W = 46, *p* = 0.031, r = 0.42). Relative sugar and fiber intake were not significantly different between the groups. However, when regarding absolute fiber intake, the cyclists and controls did not meet the guidelines for fiber intake of 14 g/1000 kcal, with an absolute intake of 26.4 ± 8.1 g (35.0 g advised) and 17.2 ± 6.8 g (20.3 g advised), respectively [13].

A qualitative analysis of the food diaries revealed distinct dietary patterns consistent with elite endurance training. The cyclists’ diets were characterized by the consumption of simple carbohydrates and protein sources around training sessions, such as energy gels and bars, sports drinks, and protein-rich recovery shakes and snacks. Although this was limited because of the “off-season”. In both groups, none of the participants reported the use of supplements or medication during and prior to the measurement period.

### 3.3. Alpha- and Beta-Diversity

The Shannon index was significantly lower in cyclists compared to the matched control group (β = 0.42, 95%CI [0.02; 0.81], Cohen’s d = 0.76), even after correction for relative fiber intake (β = 0.42, 95%CI [0.04; 0.79], Cohen’s d = 0.76). Neither richness (β = 5.2, 95%CI [−0.10; 10.5], Cohen’s d = 0.70) nor evenness (β = 0.09, 95%CI [−0.01; 0.19], Cohen’s d = 0.68) was significantly different between the two groups (see Figure 1).

Similarly To the alpha-diversity, assessment of inter-individual variability (i.e., Bray–Curtis and Jaccard Index) via Adonis PERMANOVA revealed significant separations or clustering between both groups (F = 4.429, *p* = 0.007, R^2^ = 0.15), indicating that bacterial species were clustered within the cyclists (see Figure 2).

### 3.4. Phylum Level

On the phylum level, the gut microbiome of the cyclists is dominated by Bacteroidota (72.7%) with a lower relative abundance of Firmicutes (22.1%). This is the opposite of the control group, where the microbiome is dominated by Firmicutes (62.5%) over Bacteroidota (15.3%) (see Figure 3). So the relative abundance of Bacteroidota is significantly higher in female elite cyclists (log_2_FC = 2.2, *p* < 0.001, r = 0.70), while the relative abundance of Firmicutes is significantly higher in the control group (log_2_FC = −1.5, *p* < 0.001, r = 0.67). Actinobacteriota were relatively higher abundant in the control group (6.8%) compared to the female cyclists (0.2%) (log_2_FC = −5.2, *p* < 0.001, r = 0.82).

### 3.5. Family Level

Differential abundance analysis identified 10 bacterial families with significantly different relative abundances between cyclists and controls. Within the phylum *Actinobacteriota, Eggerthellaceae* (log_2_FC = −4.3, *p* < 0.001, r = 0.71), Coriobacteriaceae (log_2_FC = −5.7, *p* < 0.001, r = 0.72), and *Bifidobacteriaceae* (log_2_FC = −5.8, *p* = 0.003, r = 0.58) were significantly enriched in the control group. For the Firmicutes phylum, the following families were significantly lower in cyclists: *Peptostreptococcaceae* (log_2_FC = −5.7, *p* < 0.001, r = 0.71), *Lachnospiraceae* (log_2_FC = −2.3, *p* = 0.001, r = 0.62), *Bacillaceae* (log_2_FC = −2.7, *p* = 0.001, r = 0.61), *Erysipelotrichaceae* (log_2_FC = −3.1, *p* = 0.002, r = 0.58), *Ruminococcaceae* (log_2_FC = −1.5, *p* = 0.015, r = 0.47), *Anaerovoracaceae* (log_2_FC = −2.5, *p* = 0.030, r = 0.58), and *Coprostanoligenes group* (log_2_FC = −1.7, *p* = 0.039, r = 0.40). These significant families are highlighted in red in the volcano plot (Figure 4). No significant differences were found for families within the Bacteroidota phylum.

### 3.6. SCFAs

There were no statistically significant differences in fecal SCFA concentrations between the cyclists and controls (Figure 5). Mean acetate levels were 57.4 ± 23.5 µmol/g in the cyclists and 53.7 ± 24.5 µmol/g in the controls (β = 3.8, 95%CI [−14.3; 21.9], Cohen’s d = 0.16). Similarly, propionate concentrations were 17.5 ± 6.2 µmol/g vs. 16.3 ± 7.2 µmol/g (β = 1.3, 95%CI [−4.0; 6.5], Cohen’s d = 0.19), and butyrate levels were 20.7 ± 12.4 µmol/g vs. 16.8 ± 8.7 µmol/g (β = 3.9, 95%CI [−4.6; 12.4], Cohen’s d = 0.36) in the cyclists and controls, respectively.

## 4. Discussion

The results of this study demonstrated clear differences in the composition of the gut microbiome of female World Tour cyclists compared to non-athletes, characterized by significant enrichment of Bacteroidota and depletion of fiber-dependent fermenters such as *Lachnospiraceae* and *Ruminococcaceae*. These findings are in line with the hypothesis that the specific nutritional and physiological demands of elite endurance cycling shape specific gut microbial profiles. It remains unclear whether the observed microbial shifts are primarily driven by training load, dietary strategies, or a combination of both. Our findings represent a group-level signature, but the functional consequences of this “performance-adapted” profile likely vary between athletes based on their unique baseline microbiome and metabolic phenotype [40,41]. To our knowledge, this is the first study in elite female cyclists to confirm this hypothesis.

The gut microbiome of the elite female cyclists was dominated by Bacteroidota (72.7%), while the control group had a higher abundance of Firmicutes (62.5%). This shift is consistent with previous research highlighting the impact of high-carbohydrate, low-fiber diets on gut microbiota composition [13]. While the cyclists’ overall higher caloric (2499 kcal vs. 1453 kcal in controls) and protein intake (17.6% vs. 14.8%) is typical for endurance athletes, the significant depletion of fiber-fermenting taxa (e.g., *Lachnospiraceae* and *Ruminococcaceae*) points to a specific reduction in complex carbohydrates and resistant starches, a common strategy to optimize glycogen availability and minimize gastrointestinal distress during competition [29,32]. However, such diets may also contribute to systemic metabolic acidosis, as highlighted by Álvarez-Herms (2024), with effects on muscle metabolism, bone health, inflammation, and gut health. During races and recovery, their diet is even richer in simple sugars and lower in fiber, potentially favoring the growth of Bacteroidota. Furthermore, those are often delivered via ultra-processed foods (UPFs). The regular consumption of these products is increasingly implicated in gut mucosa injuries, increased intestinal permeability and reduced microbial diversity [42]. Consequently, the “high-Bacteroides, low-diversity” profile may not only be an adaptation for short-term energy harvesting but also a reflection of a diet high in UPFs that prioritizes short-term performance, potentially at the expense of long-term gut and systemic health [42]. It is important to note that our 2-day food diary (following off-season) may not fully reflect the habitual intensity of this dietary pattern, and future studies incorporating longer tracking or biomarker analysis could provide deeper insights. Furthermore, our study design cannot disentangle the specific effects of chronic UPF intake from other confounding factors, highlighting a critical area for future research.

Martin et al. (2025) [27] further highlighted that gut microbiota functionality was changed in individuals with high exercise capacity, favoring energy production from available nutrients over the degradation of complex molecules. Despite the decrease in classic butyrate producers in these female cyclists, fecal SCFA levels (i.e., acetate, propionate, and butyrate) did not differ significantly between the cyclists and matched controls, suggesting metabolic flexibility in the gut microbiome of cyclists. The cyclists’ higher habitual simple carbohydrate intake likely boosted fermentation by Bacteroidota, which produces acetate and propionate via the succinate pathway. Moreover, remaining taxa may have upregulated SCFA production to meet the demand, as seen in high-carbohydrate diets [13]. Important to note is that endurance exercise accelerates gut motility, potentially reducing SCFA absorption in the gut and therefore increasing their fecal excretion [43]. To clarify this, measuring plasma SCFA levels or assessing colonocyte health could provide insights into absorption efficiency. Furthermore, metagenomic profiling could help uncover functional pathways (e.g., SCFA production) beyond taxonomic shifts. However, other studies report similar findings, noting that athletes with very high exercise capacity exhibit higher fecal SCFA levels despite reduced microbial diversity [27,44]. Their fecal microbiota transplantation experiments in mice demonstrate that the gut microbiota from elite athletes enhances insulin sensitivity and muscle glycogen storage, highlighting the functional adaptability of the gut microbiome in response to exercise.

Lastly, the results of this study are in contrast with studies that found higher abundances of *Prevotella* or *Veillonella* in endurance athletes [21], but align with findings in cyclists and athletes consuming low-fiber diets [13]. This ambiguity may reflect sport-specific demands, adaptations, and physiology. Other studies support this ambiguity, showing that gut microbiome composition varies with the type and intensity of physical activity, independent of diet [45]. Therefore, future research should focus on longitudinal monitoring across different training and race periods to assess microbiome resilience and potential links to performance metrics. The current study took samples as close as practically possible to the off-season of the cyclists to have measurements during a “relative” untrained status. However, most riders already restarted their training on the bicycle (13.3 ± 4.1 h/week), which is still relatively low compared to a typical training camp (22.5 ± 1.86 h/week). Nevertheless, these results and the literature suggest larger effects of a long-term performance-oriented lifestyle on the gut microbiome, compared to the smaller effects of acute changes in exercise and nutrition [46].

This study focuses on a unique and understudied population of elite female World Tour cyclists. The use of well-matched controls and the combination of 16S rRNA gene sequencing with SCFA quantification to link taxonomy with function makes this an insightful characterization of the gut microbiome in elite female cyclists with important hypotheses for future research. The sample size, while comparable to other studies investigating unique elite athlete populations, is a limitation, and findings should be interpreted with caution. Future multi-team studies with larger cohorts are needed to confirm these results. However, the cross-sectional design can identify an association but cannot establish causality between the performance-oriented lifestyle and the observed microbiome profile. Secondly, the current analysis was limited to taxonomic profiling via 16S rRNA gene sequencing, while metagenomic or metatranscriptomic approaches would be required to confirm the functional metabolic pathways that are hypothesized (e.g., succinate pathway upregulation).

## 5. Conclusions

In conclusion, elite female World Tour cyclists exhibited a gut microbiome that seems adapted to the nutritional and physiological demands of high-level endurance sport. This microbial profile, characterized by Bacteroidota dominance and reduced diversity, likely facilitates rapid energy harvesting from a diet high in simple carbohydrates and UPFs. For some, this adaptation may best support energy extraction and metabolic resilience. For others, it might predispose them to gut issues or suboptimal recovery. Given the cross-sectional design, no causal conclusions can be drawn. Future longitudinal studies should explore how training phases, dietary periodization, and targeted nutritional interventions (e.g., fiber reintroduction and/or personalized nutritional strategies that consider an athlete’s unique gut microbiome) influence microbiome dynamics and their relationship with performance and health outcomes.

## Figures and Tables

**Figure 1 microorganisms-13-02345-f001:**
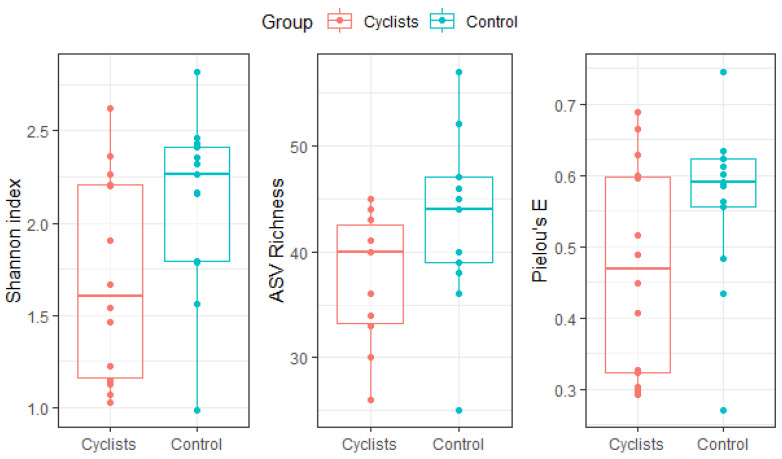
Gut microbiome α-diversity (i.e., Shannon index), ASV richness and evenness (i.e., Pielou’s E) for both groups (i.e., cyclists, controls). Dots represent individual values.

**Figure 2 microorganisms-13-02345-f002:**
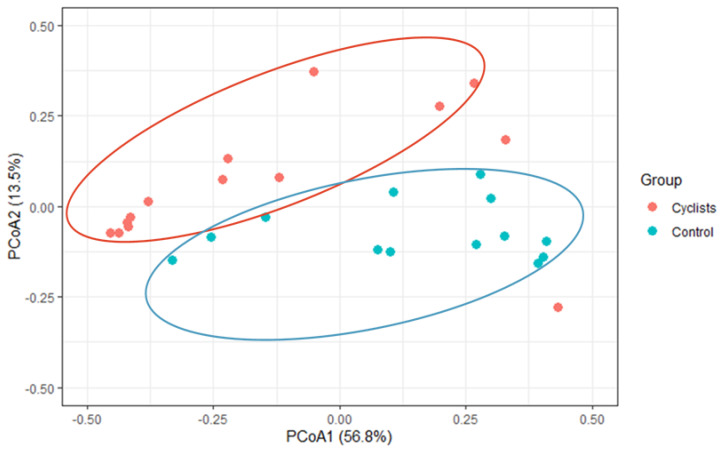
β-diversity (Bray–Curtis) via principal coordinate analysis (PCoA). A significant clustering within the cyclist and control group, respectively, was found (*p* = 0.007).

**Figure 3 microorganisms-13-02345-f003:**
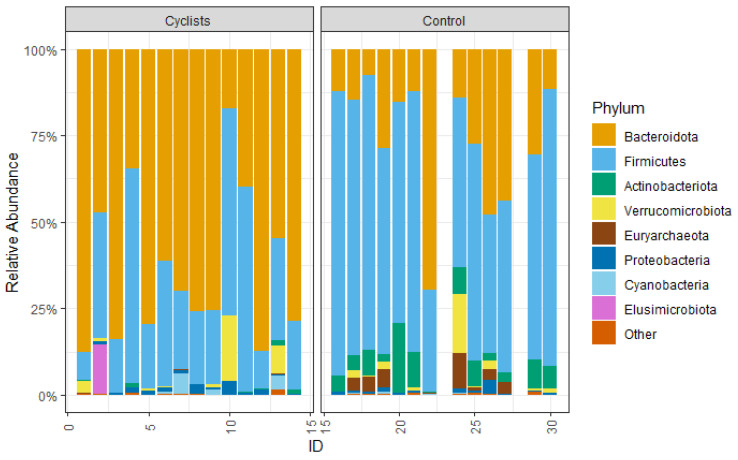
The relative abundance of the measured phyla.

**Figure 4 microorganisms-13-02345-f004:**
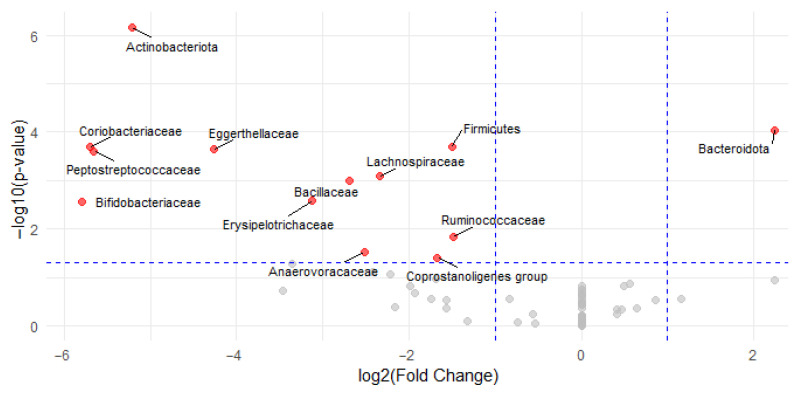
Volcano plot of differential bacterial family abundances between cyclists and controls. Each point represents a bacterial family, plotted by log_2_FC (*x*-axis) and −log10(*p*-value) (*y*-axis). Red points indicate families with statistically significant differences in abundance (*p* < 0.05 and |log_2_FC| > 1); gray points are not significant. Families enriched in the control group are shown on the left (negative log_2_FC) and in the cyclists on the right (positive log_2_FC).

**Figure 5 microorganisms-13-02345-f005:**
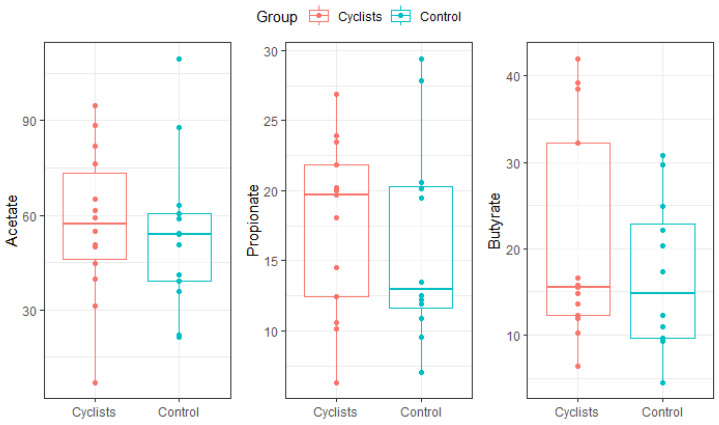
SCFA concentrations (µmol/g) for both groups (i.e., cyclists, control). Dots represent individual values.

**Table 1 microorganisms-13-02345-t001:** Diet and gastrointestinal symptoms.

	Cyclists (n = 14)	Controls (n = 13)	
	Median [IQR]	Median [IQR]	*p*-Value
Energy intake (kcal)	2499 [187]	1453 [660]	0.001
Protein (%)	17.6 [3.2]	14.8 [2.0]	0.004
Fat (%)	27.2 [4.3]	32.7 [8.0]	0.094
Carbohydrate (%)	51.1 [7.5]	49.1 [8.8]	0.275
Starches (%)	45.9 [13.9]	66.3 [24.1]	0.031
Sugar (%)	35.8 [5.1]	33.5 [18.8]	0.396
Fibers (%)	9.2 [1.8]	8.8 [3.7]	0.645
GISS (0–10)	2.0 [1.7]	2.1 [1.5]	0.542
BSC (1–7)	3 [2]	3 [4]	0.551

IQR = interquartile range; GISS = gastrointestinal symptoms; BSC = Bristol Stool Chart.

## Data Availability

The original contributions presented in this study are included in the article. Further inquiries can be directed to the corresponding author.

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
