# Peer review of "Nutritional and Physiological Demands Shape the Gut Microbiome of Female World Tour Cyclists"

_microorganisms, 2025, doi:10.3390/microorganisms13102345_

Round 1
Reviewer 1 Report
Comments and Suggestions for Authors
The manuscript presents a cross-sectional study analysing the gut microbiome of elite female World Tour cyclists compared to a control group of non-athlete females matched for age and BMI. The cyclists show significant changes in microbiome composition, with a predominance of Bacteroidota and a reduction in Firmicutes compared to the controls, as well as lower alpha diversity. The authors suggest that this profile may represent a functional adaptation of the microbiome to the nutritional and physiological demands of high-level endurance.
- Line 77, I recommend that the authors indicate what is meant by the acronym IOC.
- The introduction is well structured and comprehensive but at times lengthy and detailed; it could benefit from being more concise, focusing immediately on the specific importance of the microbiome in elite female cyclists.
- I recommend that authors also briefly explain in the final part of the introduction what experiments and tests were carried out, in order to make the aim clearer. It would be useful to emphasise more strongly the novelty of the study, given the scarcity of literature on elite female athletes, in order to better justify the work.
- The sample size (15 cyclists initially) is quite limited for a microbiome study with high individual variability. I would advise the authors to highlight the sample size limitations and the impact on statistical power.
- The control group is matched for age and BMI but not for other potential confounding variables (e.g. lifestyle, usual diet, use of medication); I advise the authors to discuss this.
- Since diet is one of the key factors in modulating the microbiome, a more detailed description of the dietary approaches and supplements used would be advisable.
- I advise authors to include an image or table summarising the data presented in paragraphs 3.1 and 3.2 in order to make the manuscript clearer and more immediately impactful.
- Indicate in the main text which figure the data description refers to.
- Looking only at Fig. 5, it is not clear which are the main bacterial families in the group of cyclists and those in the control group. I would advise the authors to make this figure clearer.
- I recommend adding an image for paragraph 3.6 as well, even though there are no statistical differences, this is still an interesting data.
- Suggestion: It would be beneficial to propose or outline possible nutritional interventions to be tested in order to modulate the microbiome of athletes in a beneficial manner.
Author Response
Response to Reviewer 1
We sincerely thank the reviewers for their time, insightful comments, and constructive criticism. Their feedback has been invaluable in strengthening our manuscript. We have carefully considered each point and have revised the manuscript accordingly. All changes have been highlighted in the revised manuscript. Our point-by-point responses are detailed below.
- Line 77, I recommend that the authors indicate what is meant by the acronym IOC.
We agree. The acronym has been defined on first use in the introduction (Line 72): "...the International Olympic Committee (IOC) consensus...".
- The introduction is well structured and comprehensive but at times lengthy and detailed; it could benefit from being more concise, focusing immediately on the specific importance of the microbiome in elite female cyclists.
We have shortened the introduction to be more focused and have explicitly highlighted the novelty of studying elite female cyclists in the final paragraph of the introduction. Adaptations are highlighted within the improved introduction.
- I recommend that authors also briefly explain in the final part of the introduction what experiments and tests were carried out, in order to make the aim clearer. It would be useful to emphasise more strongly the novelty of the study, given the scarcity of literature on elite female athletes, in order to better justify the work.
We have added a sentence to the end of the introduction to better outline the methods (Line 95): "This study addresses this gap by profiling the gut microbiome of elite female cyclists com-peting at the World Tour level. It is hypothesized that their gut microbiome has evolved due to the chronic physical stressor of cycling at the highest level, as well as the specific diet and supplementation used to optimize their endurance performance. Therefore, to our knowledge, this is the first study to comprehensively profile the gut microbiome, using both 16S rRNA sequencing and SCFA quantification, in elite female World Tour cyclists during the off-season, comparing them to well-matched non-athlete controls."
- The sample size (15 cyclists initially) is quite limited for a microbiome study with high individual variability. I would advise the authors to highlight the sample size limitations and the impact on statistical power.
This is a valid point. We have added a sentence to the discussion acknowledging this limitation (Line 385): "The sample size, while comparable to other studies investigating unique elite athlete populations, is a limitation and findings should be interpreted with caution. Future multi-team studies with larger cohorts are needed to confirm these results."
- The control group is matched for age and BMI but not for other potential confounding variables (e.g. lifestyle, usual diet, use of medication); I advise the authors to discuss this.
We acknowledge the influence of other potential confounding factors. The control group was primarily matched for age and BMI, however all other inclusion and exclusion criteria were applied on this participants as well (see list below). We extended the summary of those criteria that were mentioned in the method section (Line 107) to be more clear on this point. A difference in exercise and diet was expected and intended, since we hypothesized: ‘It is hypothesized that their gut microbiome has evolved due to the chronic physical stressor of cycling at the highest level, as well as the specific diet and supplementation used to optimize their endurance performance.’. Furthermore, none of the participants (in both groups) reported the use of medication, which was added to the results section (Line 256): “In both groups, none of the participants reported the use of supplements/medication during and prior to the measurement period.”
(Injury of long duration (>1month) at baseline; Illness (>1 week ) at baseline; Antibiotic use <14 days to baseline; Previous myocardial infarction and/or previous supraventricular/ventricular arrhythmia; Known inflammatory bowel disorder; Known intestinal motility disorder; Alcohol (defined as more than 14 units per week) or other substance abuse; Smoking; Known systemic or auto-immune disorder with implications for the GI system; Prior abdominal surgery (except appendectomy or cholecystectomy more than 6 months ago); Any prior diagnosis of cancer other than basocellular carcinoma; History of gastro-enteritis in the past 8 weeks. Defined as an acute infection of the gastrointestinal wall, with symptoms such as cramping, abdominal pain, watery, sometimes bloody, diarrhoea nausea and vomiting, lasting more than 1 day; Treatment with neuromodulators (one neuromodulator taken at a stable dose for more than 12 weeks is allowed); Treatment with spasmolytic agents, opioids, loperamide, gelatine tannate or mucoprotectants during the past 8 weeks)
- Since diet is one of the key factors in modulating the microbiome, a more detailed description of the dietary approaches and supplements used would be advisable.
Diet is indeed a key factor in modulating the gut, so we added an extra part in the results, after the quantitative analysis (Line 252): “A qualitative analysis of the food diaries revealed distinct dietary patterns consistent with elite endurance training. The cyclists' diets were characterized by consumption of simple carbohydrate and protein sources around training sessions, such as energy gels and bars, sports drinks and protein-rich recovery shakes and snacks. Although this was limited because of the “off-season”. In both groups, none of the participants reported the use of supplements/medication during and prior to the measurement period.
- I advise authors to include an image or table summarising the data presented in paragraphs 3.1 and 3.2 in order to make the manuscript clearer and more immediately impactful.”
Indicate in the main text which figure the data description refers to.
We have created a new Table 1 summarizing all dietary and GI symptom data. We have also ensured that all figure references in the text (e.g. Figure 2, Figure 3) are correct and correspond to the newly numbered figures.
- Looking only at Fig. 5, it is not clear which are the main bacterial families in the group of cyclists and those in the control group. I would advise the authors to make this figure clearer.
We have improved the legend for Figure 5 (now Figure 4) to clarify that all significantly different families are depleted in cyclists: “Families enriched in the control group are shown on the left (negative log2FC) and in the cyclists on the right (positive log2FC).”
- I recommend adding an image for paragraph 3.6 as well, even though there are no statistical differences, this is still an interesting data.
As suggested, we have created a new Figure 5 presenting the SCFA concentration data.
- Suggestion: It would be beneficial to propose or outline possible nutritional interventions to be tested in order to modulate the microbiome of athletes in a beneficial manner.
This is an excellent suggestion. We have added a sentence to the conclusion (Line 399): "Future longitudinal studies should investigate how training phases, dietary periodization and targeted nutritional interventions (e.g. fibre reintroduction or prebiotics during the off-season or between races, multi-strain pro-biotics, etc.) influence microbiome dynamics and their relationship with both performance and health outcomes."
Reviewer 2 Report
Comments and Suggestions for Authors
This cross-sectional study examines the gut microbiome differences between elite female World Tour cyclists and non-athletes and links this difference to the different physiological and nutritional requirements of endurance cycling. Authors are advised to make the following changes to the manuscript:
- Lines 21-23: Please rephrase, adjusting the language and tenses. Also, clarify the meaning of "off-season," e.g., "non-competition times."
- In the introduction section, please explain why the study was conducted on females and not males.
- The age of the control group was determined. What about the ages of the female athletes?
- Do the authors believe that a two-day food diary, during the off-season, is sufficient to judge on dietary requirements and habits? Would two days have an impact on the gut microbiome? Why, for example, was a food frequency questionnaire not conducted?
- The food frequency questionnaire would also have provided insights into the intake of various types of foods, including fiber-rich vegetables and fruits, and dairy products, in addition to the extent to which the players adhered to certain dietary patterns, especially off-season.
- The subheadings "3.1. Descriptives" as well as "3.2. Diet and Gastro-Intestinal Symptoms", The information under each of them should preferably be placed in a table for clarity of presentation.
- Line 264, figure 2 or 1?
- Subheading "3.6. SCFA", please put the data below in a table.
- Please check the tenses and language carefully.

Author Response
We sincerely thank the reviewers for their time, insightful comments, and constructive criticism. Their feedback has been invaluable in strengthening our manuscript. We have carefully considered each point and have revised the manuscript accordingly. All changes have been highlighted in the revised manuscript. Our point-by-point responses are detailed below.
- Lines 21-23: Please rephrase, adjusting the language and tenses. Also, clarify the meaning of "off-season," e.g., "non-competition times."
We have rephrased the relevant sentences in the abstract for clarity and defined the sampling period as "a period of reduced training (off-season)."
2. In the introduction section, please explain why the study was conducted on females and not males.
We have shortened the introduction to be more focused and have explicitly highlighted the novelty of studying elite female cyclists in the final paragraph of the introduction (Line 93-102).
3. The age of the control group was determined. What about the ages of the female athletes?
This was not clearly mentioned in the methods section of the paper, but for the cyclists there was no inclusion criteria based on age. Nevertheless, cyclists of a World Tour team cannot be younger than 18 years old. The upper age-limited was, on the other hand, based on the age of the oldest participating cyclist (34 years old). This was clarified in the methods section (Line 111): “The control group (n=15) consisted of age-matched (i.e. age range of the cyclists: 18–34 years) and BMI-matched (<25 kg/m²) females.”.
4. Do the authors believe that a two-day food diary, during the off-season, is sufficient to judge on dietary requirements and habits? Would two days have an impact on the gut microbiome? Why, for example, was a food frequency questionnaire not conducted?
The food frequency questionnaire would also have provided insights into the intake of various types of foods, including fibre-rich vegetables and fruits, and dairy products, in addition to the extent to which the players adhered to certain dietary patterns, especially off-season.
We have added a justification for the 2-day diary in the methods section, acknowledging its limitations. We agree that an FFQ would provide valuable insights into long-term habits. We have mentioned this as a recommendation for future research in the discussion. Unfortunately, we were strictly limited in the amount of time that we could take of the riders. The study is also part of a larger study, where we follow up the riders for a whole season, so a 2-day diary is in practical terms more usable for this purpose (Line 132).
“A two-day food diary was chosen to minimize participant burden on the elite athletes while still capturing recent dietary intake, acknowledging that it may not fully represent long-term habitual diet. Future studies would benefit from longer tracking periods or food frequency questionnaires.”
We acknowledge this limitation in the discussion as well (Line 336):
“Although these differences in gut microbiome align with existing literature, the 2-day food diary (following off-season) may not fully reflect habitual dietary intake. Extending the tracking period or incorporating biomarker analysis (e.g. faecal starch) could provide deeper insights into fibre’s role.”
5. The subheadings "3.1. Descriptives" as well as "3.2. Diet and Gastro-Intestinal Symptoms", The information under each of them should preferably be placed in a table for clarity of presentation. Line 264, figure 2 or 1?
We have created a new Table 1 summarizing all dietary and GI symptom data. We have also ensured that all figure references in the text (e.g. Figure 2, Figure 3) are correct and correspond to the newly numbered figures.
6. Subheading "3.6. SCFA", please put the data below in a table.
As suggested by both reviewers, we have created a new Figure 5 presenting the SCFA concentration data.
7. Please check the tenses and language carefully.
The manuscript has been carefully proofread by a native English speaker to improve language and tense consistency.
Reviewer 3 Report
Comments and Suggestions for Authors
I have reviewed the article titled "Nutritional and Physiological Demands Shape the Gut Microbiome of Female World Tour Cyclists" and I have found some major flaws that couldn’t be rectified during revision and should be rejected. “Microorganisms” is a respected journal in MDPI, publishing such data will degrade the journal quality. Incomplete alpha and beta, along with only phylum level microbial analysis is not sufficient to push an article. Both alpha and β-diversity (Bray–Curtis) via principal coordinate analysis (PCoA) charts are completely wrong and misleading.. Figure 1. is absent in the draft, while the mentioned figures are misleading…
The title is quite interesting however the data analysis and presentation is very poor. The functional analysis (short-chain fatty acid (SCFA) mentioned in the abstract is absent in the results, conclusion section. Overall/suggestions. The authors fail to analyzed and write a well-organized article. The discussion section is completely misleading. Publishing such data will definitely disgrace the journal quality. Therefore, the article is not suitable for publication in “Microorganisms” and should be rejected.
Author Response
We thank Reviewer 3 for their time and critique. We believe the major concerns came from misunderstandings that we have now clarified in the revised manuscript.
- Comment: "Incomplete alpha and beta, along with only phylum level microbial analysis is not sufficient... SCFA analysis is absent... Figures are misleading."
- Response: We respectfully disagree with these assessments and have made significant revisions to improve clarity.
- Analysis Depth: Our analysis was not limited to the phylum level. Section 3.5 and Figure 5 (now Figure 4) present a detailed differential abundance analysis at the family level, identifying 10 significantly different families. We have made this clearer in the text and figures.
- SCFA Data: The SCFA results were already fully presented in Section 3.6. We have now also summarized this data in a new Figure 5 to enhance visibility.
- Figure Issues: The absence of "Figure 1" was actually a numbering issue. This has been corrected and all figures are now correctly numbered and referenced (Figure 1: Alpha-diversity, Figure 2: Beta-diversity PCoA, Figure 3: Phylum abundance, Figure 4: Volcano plot of family-level differences, Figure 5: SCFAs). The alpha- and beta-diversity figures are standard and correctly represent our statistical findings (significant difference in Shannon index, significant clustering via PERMANOVA).
- Discussion: We have revised the discussion to ensure it accurately reflects our results and avoids overinterpretation, focusing on the observed adaptations and their potential implications. As well as further reflecting on the limitations of our study.
We are confident that the revisions have addressed the core of the criticism and significantly improved the manuscript's quality and clarity.
Round 2
Reviewer 2 Report
Comments and Suggestions for Authors
The manuscript has been revised according to the proposed amendments.
Author Response
/
Reviewer 3 Report
Comments and Suggestions for Authors
The revision is not satisfactory...
Author Response
/